# Rapid health impact assessment of COVID-19 on families with children with disabilities living in low-income communities in Lusaka, Zambia

**Mary O. Hearst**[1☯]*, **Lauren Hughey**[2☯], **Jamie Magoon**[1‡], **Elizabeth Mubukwanu**[3‡], **Mulemba Ndonji**[3☯], **Esther Ngulube**[3‡], **Zeina Makhoul**[2☯]

**1** Public Health Department, Henrietta Schmoll School of Health, St. Catherine University, St. Paul, Minnesota, United States of America, **2** SPOON, Portland, Oregon, United States of America, **3** Catholic Medical Mission Board, Lusaka, Zambia

☯ These authors contributed equally to this work.
‡ These authors also contributed equally to this work.
* mohearst@stkate.edu

**Data Availability Statement:** Data and surveys were provided in the supplementary materials in Excel files.

## Abstract

### Introduction

Worldwide, children with disabilities are a vulnerable population and at high risk for COVID-19 morbidity and mortality. There is little information on the impact that COVID-19 had on children with disabilities and their families, particularly in low-income settings. This assessment describes the extent to which the pandemic impacted seven indicators of well-being in three low-income communities in Lusaka, Zambia.

### Methods

Interviews were conducted with a random selection of families participating in an existing program (n = 39), community health workers (n = 6), healthcare workers (n = 7) and government officials (n = 2). Descriptive data was summarized and qualitative responses reviewed for themes.

### Results

Most families reported a major loss of income resulting in food insecurity (79%), housing instability (67%), stress (36%), and increased risk of child separation and neglect (18%). Most families did not report receiving governmental financial assistance and reported loss of access to health services for their child such as physiotherapy (33%). Stakeholders interviewed reported that COVID-19 information was widely available although few specific interventions for children. Families were seen to have greater food insecurity, more poverty, more crowding, less healthcare services and children left alone or on the streets to beg.

### Discussion

COVID-19 and related containment measures have impacted the lives of children with disabilities and their families to a great extent. There is an urgent need for disability-inclusive

**Funding:** MH and ZM: GHR Foundation, no number http://www.ghrfoundation.org/ The funders had no role in study design, data collection and analysis, decision to publish, or preparation of the manuscript.

**Competing interests:** The authors have declared that no competing interests exist.

responses that deliberately address the needs of children with disabilities and their families, notably uninterrupted access to adequate food, inclusive education, rehabilitation therapy, and income-generating activities.

## Introduction

Children with disabilities and their families living in low- and middle-income countries (LMICs) encounter cumulative hardships without the presence of a global pandemic. In Zambia, an estimated 4.4% of children between the ages of 2 and 17 years have a disability [1]. Children with disabilities are a vulnerable and marginalized population experiencing conditions that increase their risk for COVID-19, including poverty, limited access to healthcare, and fewer educational opportunities [2–7]. UNICEF issued a statement highlighting that children with disabilities are highly vulnerable to being adversely affected by COVID-19 [8]. Many children with disabilities have underlying health conditions. Those with feeding difficulties, such as difficulty swallowing and sitting upright for feeding, often have increased nutritional needs and weakened immune systems [9, 10]. They frequently experience cough, aspiration, and respiratory illness [11]. Not only can these conditions be worsened by COVID-19, they may also increase the risk of serious complications from the infection. Factors compounding this risk are shelter-in-place orders, potential reduced access to health services, food insecurity, and school closures. These amplify existing hardships and increase the risks of morbidity and mortality among children with disabilities and their families. While global health leaders recognize that children with disabilities will likely be adversely impacted by COVID-19 [12, 13], it is not yet known how exactly and to what extent. As governments and civil society continue to adapt their policies and programs in response to the pandemic, new strains, and vaccine distribution, it is critical to use information from the lived experience of children with disabilities and their families to combat existing inequities and inform policy solutions for this population of children. The pandemic is ongoing and mutating; therefore, the voices from the community can inform current COVID-19 pandemic policy and future pandemics.

The purpose of this paper is to describe the perceived impacts of COVID-19 for children with disabilities and their families living in Lusaka, Zambia from the parents, community caregivers (CCGs), health professionals, and government officials.

## Materials and methods

This research was based in the capital city of Lusaka, Zambia, located in sub-Saharan Africa. The first cases of COVID-19 reached Zambia in March 2020 [14]. Zambia has experienced three distinct and accelerating waves of COVID-19 between March 2020 –July 2021 [15]. The government instituted precautionary measures including masking, limiting social gatherings and shelter-in-place during each wave (moh.gov.zm).

Catholic Medical Mission Board (CMMB) is an international non-governmental organization (INGO) with offices and programming in five countries (www.cmmb.org). The Lusaka, Zambia office has been working in partnership with St. Catherine University, St. Paul, Minnesota USA, since 2015 to build a program focusing on children with disabilities living in home-based settings. The result was Kusamala+ ("to care" in the local language of Nyanja), a CMMB-St. Catherine's joint program that increases local capacity to decrease stigma toward children with disability and their families; strengthens healthcare systems around disability; and build home and community-based social welfare and support programs [14]. Kusamala

+ is located in three low-income, high density, low-resource communities in Lusaka, Zambia. 574 children with disabilities have been identified by Kusamala+ and 522 are currently active in the program. Additional details of Kusamala+ can be found here "Community-based intervention to reduce stigma for children with disabilities in Lusaka, Zambia: a pilot" here (https://pubmed.ncbi.nlm.nih.gov/33053312/) [14].

## Participants

Using the roster of enrolled and engaged families with Kusamala+, the primary caregivers who had children with moderate to severe disabilities, as identified by the Zambian Association of Persons with Disability (ZAPD), were eligible for selection (n = 168). Twenty primary caregivers were randomly selected from the list of eligible participants. 10–15 families from each community were contacted by phone or home visits for those without phones by CMMB Kusamala+ staff, asked about their interest in participating. All of the adult caregivers in the families who were asked consented verbally to an in-depth interview with a goal of 12 families per community. No minors were interviewed and verbal consent was witnessed by the CCG working with the family and marked as "Consent" on the survey form.

As part of Kusamala+ programming, families have an assigned CCG, a community volunteer trained to provide support, referral, and education at the household level. Of the 54 active CCGs, six CCGs were purposively selected based on commitment to the program activities, were contacted by phone, and consented in a similar manner. All CCGs contacted consented. The CCGs report to healthcare professionals (nutritionist, physiotherapist, nurses, psychosocial counselor, and environmental health technician) at the local health facilities in the communities. Six healthcare professionals were randomly selected, all agreed to participate, and consented verbally similarly. The health care professionals included two environmental health technicians, one nurse, two psychosocial counselors, and two physiotherapists. Government officials from ZAPD and a provincial social welfare officer from the Ministry of Community Development and Social Welfare who were likely to provide rich information on our topics of interest were selected in order to effectively use limited resources. They represent the stakeholders as well as ministries attentive to disability. This was deemed exempt by St. Catherine University IRB and the University of Zambia IRB.

## Interview process

The interviews were semi-structured with categorical (survey-based) and open-ended questions. Interviewers were provided with a guide on how to prompt participants on open-ended questions related to how COVID-19 has impacted their lives and communities. Data collection occurred in October and November 2020 by CMMB Kusamala+ staff multi-lingual in English, Nyanja (primary language in Lusaka) and other local dialects who have extensive experience in data collection with families who have a child with a disability. CMMB data collectors were additionally trained on the study objectives, survey questions, and data entry and provided feedback on interview questions and process. Interviews were conducted in English or Nyanja over the phone or, only when necessary, in person following COVID-19 safety precautions. Interviews were not recorded. Instead, data collectors marked responses on paper forms and then transferred de-identified data to Google Forms within 48 hours. Identification numbers were assigned to participants limiting identifiability to the research team. Data collectors and the research team had access to the de-identified data.

## Measures

Interview questions were developed over a series of video conference calls between CMMB, St. Catherine and SPOON research team members. We used the Child Status Index to guide

the design of our study domains and interview questions [16]. The Framework identifies six domains of child well-being that were used as a guide for question development in the areas of 1) Food consumptions; 2) Housing and livelihood; 3) Safety and risk of separation; 4) Child health and wellness; 5) Parent stress; 6) Child stress; and 7) Education. Interview questions were organized by domain.

**Families.** Primary caregiver and child demographics existed as part of Kusamala+. The primary caregiver was asked to confirm demographic information of themselves and their child (sex, age, type of disability, educational status); knowledge of COVID-19 and experience including use, access and affordability of prevention items such as masks, soap and sanitizer; and the perceived extent to which COVID-19 impacted access to adequate food, housing and livelihood, child safety and risk of separation, child's health and wellness, parental and child stress and mental health, and education.

**CCGs and healthcare professionals.** CCGs were asked questions about their observations of the impact of COVID-19, and perceived extent to which COVID-19 impacted access to adequate food, housing and livelihood, child safety and risk of separation, child's health and wellness, parental and child stress and mental health, and education. CCGs also were asked about availability of community services for the families. Healthcare professionals were asked about practices and access to preventative materials and changes observed in healthcare delivery. Both CCGs and healthcare professionals provided recommendations. Families were not asked for recommendations directly due to the length of the survey-based interview; however, families shared gaps in services and family needs throughout.

**Government and iNGO representatives.** The two representatives were asked to respond to questions on the main issues faced by children with disabilities due to COVID-19, what programs are in place, what the gaps are, and how supporting children with disabilities can contribute to the overall recovery.

## Analysis

Anonymized data from the Google Forms were exported to a Microsoft Excel database and then quantitative components were imported into Stata v15 [StataCorp College Station TX]. Short answer responses were organized by respondent, domain, and whether or not there was a self-reported perceived impact. Descriptive characteristics of participants are presented as proportions. Braun and Clark's [17] steps in thematic analysis were used to identify patterns in themes within each HIA domain. The first and second authors read through interview transcripts several times to get a thorough understanding and overall picture of its contents. They then generated initial codes or brief descriptions of the content, which they then inputted into a database by respondent and domain. The two authors identified emerging themes within the HIA domains. The third author reviewed the transcripts, initial codes, and themes for accurate representation and discrepancies. New identified themes were added at that point. Findings were discussed and themes were defined and finalized with the local team in Zambia over a video call. Major themes are presented in this report with supporting quotes from participants.

## Results

### Families with children with disabilities (n = 39)

Ninety-two percent of the household caregiver/parent were female; 42% were 20–40 years of age, 28% 40–49 years of age and 30% older than 50 years. Seventy-one percent had primary education or less. The children with disabilities ages were 3–5 years (18%), 6–11 years (44%), 12–15 years (21%) and 16–20 years (18%); 41% were female. Sixty-four percent of children had a physical disability (e.g. cerebral palsy, injury, sickle cell anemia), 15% a cognitive

disability (Down's syndrome, unspecified), 13% loss of hearing, and 10% had both physical and cognitive delays. No family interviewed reported a confirmed COVID-19 illness in their household. However, 31% of respondents reported having a family member with symptoms. Families had access to information on COVID-19 from trusted sources (the health facility) and most used masks and handwashing (92%); 56% practiced social distancing and only 3% used hand sanitizer due to lack of availability and affordability. Families reported that masks were available (92%) and affordable (74%) but reported hand-washing supplies were less available (59% report 'available') and affordable (23% report 'affordable').

Fig 1 presents the perceived impacts that COVID-19 and related containment measures had on children with disabilities. Families reported that the pandemic greatly impacted their food consumption (79%) and housing and livelihood (67%). Examples of food impact were statements such as, "We used to have 3 to 4 meals in a day, now we have one. My child is losing weight. . ." and "We don't have access to nutrition foods, just eating the same things every time." Examples of impact on housing and livelihood include statements such as "My husband lost his job. . ."; "I am in a lot of credits but I still owe my landlord for four months."; and "Our income is reduced. . ." This, in turn, impacted their stress levels and caregiving practices, including leaving the child home alone while seeking paid work. Key health services like physiotherapy were no longer available primarily due to clinics being closed due to COVID-19. Of the few children who attended school (28%), most report schools have closed or reduced hours.

## CCG, healthcare professionals, government officials

Major themes reported by families and corroborated by community leaders, health facility staff, and government representatives are summarized in Table 1. CCGs reported that families

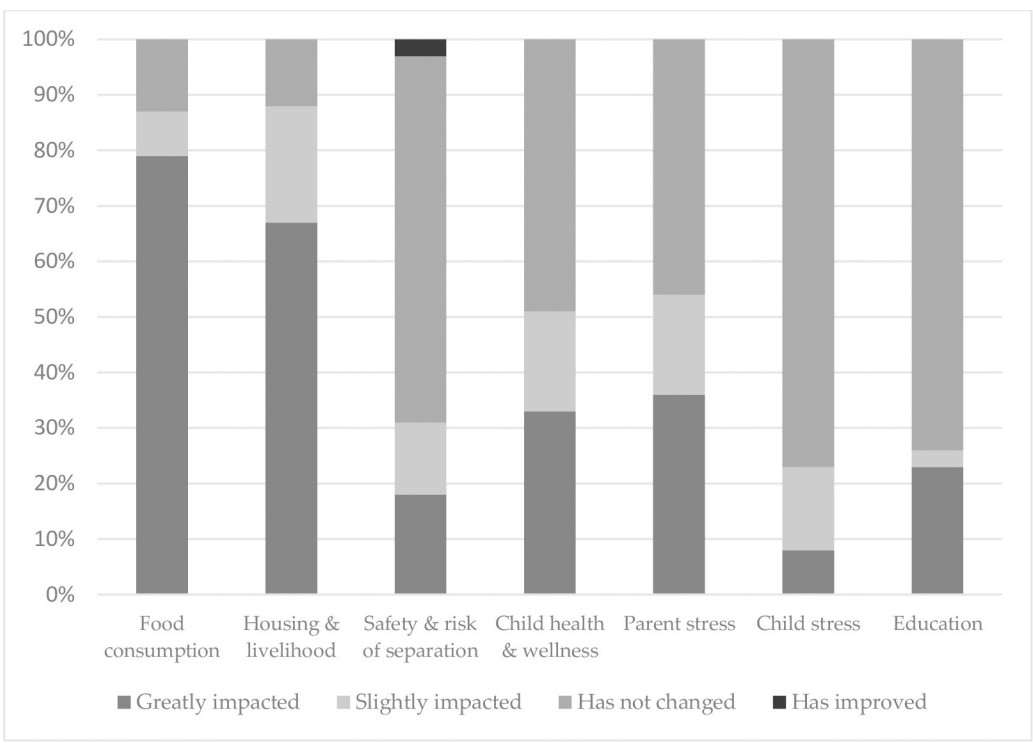

**Fig 1. Perceived negative impacts of COVID-19 on children with disabilities and their families (n = 39).**

**Table 1. Major themes around the perceived impacts of COVID-19 on children with disabilities and their families in Lusaka, Zambia.**

| Domain | Themes | Supporting quotes |
|---|---|---|
| **COVID-19 knowledge and practices** | • Community-level COVID-19 prevention messages widely shared and accessible<br>• Absence of specific interventions for children with disabilities | *I am not stressed because I know what to do to protect myself and my family from COVID-19.* (Female, 30–39 years, Chawama) |
| **Food consumption** | • Increased food insecurity (lower meal quantity and quality) | *We used to have 3 to 4 meals in a day. Now we have 1 to 2 meals in a day. My grandchild has reduced weight. I have only received 1 bag of mealie meal from some institution 3 months ago* |
| | | Female, 50–59 years, Misisi |
| **Housing and livelihood** | • Family loss of income, livelihood, and housing<br>• Renters at highest risk for housing instability<br>• Increased household crowding | *We have less income as family because they only depend on me to provide and I had lost employment during this same period of COVID-19.* |
| | | Female, 40–49 years, Misisi |
| **Child safety and risk of separation** | • Child left alone as caregivers leave the house in search of employment and food<br>• Increased child begging in the street for food | *I am not able to fully take care my child because I have to look for money so I leave him home.* |
| | | Female, 60+ years, Kanyama |
| **Child health and wellness** | • Reduced access to primary care<br>• No access to physiotherapy<br>• Unable to afford medicine | *We have been told not to go for physiotherapy because of COVID-19. At times, I delay to seek medical care when am sick fear of been exposed to COVID-19 and we usually find it difficult to access transportation.* |
| | | Female, 60+ years, Misisi |
| **Parental and child stress** | • Loss of income and food insecurity as major stressors<br>• Families coping with the help of family, community, and church | *I mostly have had headaches with a lot of questions like where to find food, diapers for my child and others and this drains me a lot.* |
| | | Female, 20–29 years, Misisi |
| **Education** | • No or limited school attendance prior to COVID-19<br>• Lack of special education resources | *He completely stopped attending school. Schools were closed.* |
| | | Female, 40–49 years, Kanyama |

experience barriers to adhering to COVID-19 prevention measures for themselves and their children. These barriers include a lack of access to hand washing facilities, soap and sanitizing products, and personal protective equipment (PPE).

> "Because of the guidelines of masking up, CWDs are not free to be in the healthy facility, they are told to stay home, they may be washing hands but not with soap, they have no access to handsanitizers." (CCG)

In addition, CCGs commonly reported that children with disabilities were unable to practice hand hygiene and wear PPE due to the nature of their disability. COVID-19 information is reported to be widely circulated and understood, but not accessible for those with hearing impairments. CCGs are able to maintain preventative practices; however, some expressed concern for their own safety and continued access to PPE.

Health care professionals report PPE supplies, soap, and COVID-19 information and guidelines are generally available at health facilities. Of these items, only COVID-19 information and guidelines were reported to be adequate. CCG supervisors reported modest availability of hand sanitizers and generally inadequate availability of hand sanitizer, PPE, and soap supplies at their facilities. Changes to safety procedures at the health facilities included an increased number of handwashing stations and social distancing measures. All respondents reported changes in increased support in the form of donated PPE supplies to the health facilities by non-governmental organizations and cessation of regular activities. However, children with disabilities do not have access to information and do not have resources to acquire PPE

supplies. Existing COVID-19 information is not adapted for children with disabilities to easily understand. There are no programs or policies in place to identify children at high risk of COVID-19 and the secondary impacts of COVID-19.

Most respondents reported changes to clinic schedules including closures and revised clinic times or frequencies; limited access to clinical care and prioritization of critical cases; and reduced number, size, or length of appointments. The reduction in clinic services and schedules negatively impacted children with disabilities and their access to health care. While children have maintained access to primary care services, access to physiotherapy was reduced or eliminated for these children. One CCG supervisor reported that these changes cause parents to delay bringing children to the health facility until they are critically ill. CCGs expressed concerns about the increased risk for children with disabilities being infected with COVID-19 and having serious health outcomes as a result.

"They are able to access health care like getting medications but they have not had physiotherapy." (CCG)

CCGs, supervisors, and government representatives concur that there was a reduction in families' food security, including a reduction in the number and nutritional value of meals. Some CCGs reported food distribution by NGOs and individuals; however, the impact of these interventions was very small as they did not reach many families with children with disabilities. A government official reported that a Social Cash Transfer system exists, but there is not enough funding to support all families. CCGs, supervisors and government representatives affirmed families' experiences with loss of income and housing insecurity. Most families had to stop or curtail their income-generating activities due to containment measures. As a result, families were forced to move, or combine households, leading to crowded living situations.

"They don't have enough food, most of them stopped working"Physiotherapist

"Some families have lost housing because they cannot afford to pay rentals." Nurse

"Too many people in a household" CCG

Half of the CCGs reported instances of child neglect among the families they serve. CCGs attributed the cause of neglect to the loss of livelihood and increased food insecurity. CCG supervisors added that the need for caregivers to look for food and employment while leaving children unattended places children with disabilities were at greater risk of family separation. Conflictingly, most CCG supervisors also reported that children with disabilities are increasingly being accepted by their communities and families. One CCG stated, "Children with disabilities are now accepted by the community and some times the community helps in identifying children with disabilities." Yet, government officials highlighted the gaps in programming for children with disabilities generally and the lack of assistive devices specifically.

"Non availability of PPEs, income at household level is very low because most parents stayed away from work, others lost their jobs and so it is very difficult to feed their families" Officer from Ministry of Community Development and Social Services

"There is no program currently or maybe I don't know." Officer from Ministry of Community Development and Social Services

There were repeated themes of food insecurity and loss of livelihood contributing to increased parental and child stress. Many families do not have adequate resources or support,

leading CCGs to fear for the health of children with disabilities. A government official highlighted an example of how the stress caused by COVID-19 can lead to family separation:

> "When a couple faces challenges in supporting the child, the couple is likely to separate from each other, hence even the children with disabilities suffer." CCG

CCGs and CCG supervisors reported low school attendance before COVID-19 containment measures were enacted. Therefore, the effect of school closures is limited as most children with disabilities do not attend school. No specific interventions focused on increasing access to education for children with disabilities during COVID-19. Government officials reported that online schools are not accessible for children with disabilities and there is a lack of facilities for special education.

Key stakeholders interviewed reported the need for interventions, such as consistent and adequate food support and cash transfers, for households with children with disabilities.

> "I think government should supply CWDs with food, Social cash transfer should be consistent." Psychosocial Counselor

> "Some children with certain disabilities face more challenges when it comes to mobility. They need more support in accessing PPEs and other services. Information should be brought to their level." Zambian Association of Disabilities Representative

## Discussion

The COVID-19 pandemic will likely have profound and long-lasting effects on all families and those with children with disabilities. Families are aware of COVID-19 prevention practices with the most common being handwashing and the use of face masks; however, social distancing was less common as overcrowding is a concern and the use of hand sanitizer and soap was limited due to supply constraints. The most dramatic reported changes have been in the areas of food consumption, housing, and livelihood. These findings are an important contribution to practice and research; to our knowledge, limited work has been published on the impacts of COVID-19 on children with disabilities living in low-income communities in Zambia.

Consistent with the findings here, persons with intellectual disabilities in Botswana lacked access to adequate access to PPE and information in a manner that was accessible to them [18]. In addition, persons with disabilities were identified to be more at risk of illness and greater difficulties maintaining social distance and hygiene given need for personal care [18, 19]. The need for PPE and access to information was a recommendation expressed by the CCGs, supervisors and government officials in the current study. Somewhat consistently, persons with disabilities were found to lack healthcare and support in South Africa due to COVID-19, including limited access to intensive care unit beds and ventilators, loss of therapeutic interventions and medication [20]. The participants of this study reported some access to regular care and medications although did note that physiotherapy was not available. Research from multiple sub-Saharan African countries also noted the loss of education opportunities for children with disabilities due to COVID-19 in addition to educational systems not meeting the needs during non-pandemic times [19–21]. Finally, children with disabilities and their families in Uganda are facing loss of employment resulting in housing and nutrition deficiencies, difficulty accessing healthcare, and stress [21]. Children aged 12–25 years who had disabilities or were disadvantaged in Zambia and Sierra Leone who had low levels of health literacy were more likely to have lower mental well-being scores; however, the study found

considerable resilience among interviewed persons [22]. The reports across sub-Saharan Africa are consistent and a directed response is needed urgently.

The reported impact on food consumption has implications for growth and development among a group of children already at risk for malnutrition and delays [15]. Children with disabilities in LMICs are three times more likely to be underweight and twice as likely to experience stunting and wasting, than children without disabilities [23]. Globally, children with disabilities and their families are at higher risk of experiencing food insecurity even before the presence of a global pandemic [24, 25]. Their risk of food insecurity is further compounded during times of crisis when access to adequate, nutrient-dense foods is limited, increasing the risk for malnutrition and negative health outcomes [24]. Further, malnutrition in concert with COVID-19 places children at greater risk of co-infections and co-morbidities [26] due to barriers in healthcare delivery and competing national health priorities.

Disability, poverty and stigma are three factors that increase risk for child abandonment [27]. The additional child safety risks and abandonment concerns reported by parents included leaving their children behind while they search for work, either with a family member or alone. The presence of food insecurity in the household, inextricably tied to poverty, contributes to the risk of children leaving home in search of food [28, 29]. These put the children's safety at risk and vulnerable to potential mistreatment and abuse by strangers or those not familiar with the child's disability. Some families also face stigma and isolation due to negative community-held beliefs about disability, compounding the impact of household food insecurity and increasing the likelihood of abandonment should families be unable to get adequate support from the community [6]. Exacerbating the risk to children, due to the wrong belief that institutional care, rather than family care, is in the best interest of the child, children with disabilities are at increased risk of separation from their families in times of crisis [27]. Furthermore, COVID-19 has resulted in over 1 million children losing a primary caregiver [30]. Although the data did not report children with disabilities in particular, the loss of a primary caregiver for a child with disabilities could be detrimental even though there is improved acceptance of children with disabilities in the community.

The reported housing insecurity and the need to combine households among participants is a concern for overcrowding and increases the risk of spreading COVID-19 to vulnerable children [31]. A reduction in access to health services and medicines reported among some families poses a threat to children with disabilities, as many of these therapies and community-based rehabilitation services such as physiotherapy offers parents and children psychosocial support and serves as an entry point to accessing other important health services. Fear of exposure to COVID-19 was a reported barrier in seeking services for their children, and families even reported delaying care only until necessary [31], potentially risking the child's health. The stress experienced by parents and children with disabilities as a result of the COVID-19 pandemic is only exacerbated by the loss of livelihood, income, crucial health services, and human interaction.

This study has important public health implications and highlights the presence of weak healthcare systems and the inequities that children with disabilities face in receiving adequate health services [32]. Specifically, systems fail to address the unique needs of children with disabilities in times of emergency. Children with disabilities are typically the last to receive emergency aid and support and are often marginalized and deprioritized over others, putting them at risk of abandonment by their families and the community [33]. This practice is a human rights violation against children with disabilities and requires action and reform. As a majority of the scientific literature surrounding the COVID-19 pandemic is focused on middle- to upper-income communities and adult or elderly populations, the public health and research community must consider the unique challenges that exist for children with disabilities in

low-resource settings [32]. This study recognizes that further research is needed to understand the disparate impact of COVID-19 and its response measures on children with disabilities compared to children without disabilities living in the same communities.

Coker, et al. argues that expanding access to child safety nets, including housing and legal protection, is an ethical imperative for governments to implement [26]. Further, a collective effort must be made by leaders across sectors including non-governmental organizations, health professionals, and other stakeholders to jointly assure the protection of children, particularly in a pandemic that yielded many unintended consequences [26]. Kusamala+ was intentionally designed to strengthen the healthcare system, strengthen families and the community regarding disability, and bridge governmental and non-governmental organization support across the sectors of health, social welfare and education. The stakeholder group, that includes these players, advocate for the rights and well-being of children with disabilities while ongoing research continues for implementation and translation.

This study has limitations that could be addressed in future research. The sample size used in the study is small. It only included families with children with disabilities and does not include a comparison group. Additionally, the population included in the study were from the Kusamala+ program who were already engaging in community-based programming. This results in sample bias because Kusamala+ families may be experiencing more social support and therefore, a different pandemic experience with their children with disabilities.

## Conclusions

Children with disabilities and their families remain a hidden and high-risk population, particularly in low-resource environments. Recommendations include assuring policies are not only inclusive of disability, but intentional toward children with disabilities. Uninterrupted access to adequate services, education, and support must be available during the pandemic and beyond so that children with disabilities have a chance to engage fully in social and civic life. While ambitious goals, these children are often hidden in society. This data provides an opportunity to illuminate the need to move children with disabilities to the forefront of decision-makers and the community of practice's agenda.

## Supporting information

**S1 Data.**
(ZIP)

## Acknowledgments

The authors acknowledge the participants who responded to interview requests, in particular the families with children with disabilities. The authors also acknowledge all partners and contributors to the larger Kusamala+ program.

## Author Contributions

**Conceptualization:** Mary O. Hearst, Lauren Hughey, Elizabeth Mubukwanu, Mulemba Ndonji, Zeina Makhoul.

**Data curation:** Mary O. Hearst, Elizabeth Mubukwanu, Mulemba Ndonji, Zeina Makhoul.

**Formal analysis:** Mary O. Hearst, Lauren Hughey, Jamie Magoon, Zeina Makhoul.

**Funding acquisition:** Mary O. Hearst, Zeina Makhoul.

**Methodology:** Mary O. Hearst, Zeina Makhoul.

**Project administration:** Mary O. Hearst, Lauren Hughey, Elizabeth Mubukwanu.

**Supervision:** Mulemba Ndonji, Zeina Makhoul.

**Writing – original draft:** Mary O. Hearst, Lauren Hughey, Zeina Makhoul.

**Writing – review & editing:** Mary O. Hearst, Lauren Hughey, Jamie Magoon, Elizabeth Mubukwanu, Mulemba Ndonji, Esther Ngulube, Zeina Makhoul.

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
