## [Decision Letter · Decision Letter 0]

10 Aug 2021

PONE-D-21-13750

Rapid Health Impact Assessment of COVID-19 on Families with Children with Disabilities Living in Three Low-Income Communities in Lusaka, Zambia

PLOS ONE

Dear Dr. Hearst,

Thank you for submitting your manuscript to PLOS ONE. After careful consideration, we feel that it has merit but does not fully meet PLOS ONE’s publication criteria as it currently stands. Therefore, we invite you to submit a revised version of the manuscript that addresses the points raised during the review process.

We look forward to receiving your revised manuscript.

Kind regards,

Maria Berghs, PhD

Academic Editor

PLOS ONE

1. Please ensure that your manuscript meets PLOS ONE's style requirements, including those for file naming. The PLOS ONE style templates can be found at https://journals.plos.org/plosone/s/file?id=wjVg/PLOSOne_formatting_sample_main_body.pdf and https://journals.plos.org/plosone/s/file?id=ba62/PLOSOne_formatting_sample_title_authors_affiliations.pdf.

4. Please include your tables as part of your main manuscript and remove the individual files. Please note that supplementary tables (should remain/ be uploaded) as separate "supporting information" files.

Additional Editor Comments (if provided):

Reviewers' comments:

Reviewer's Responses to Questions

**Comments to the Author**

1. Is the manuscript technically sound, and do the data support the conclusions?

Reviewer #1: No

Reviewer #2: Yes

Reviewer #3: Partly

2. Has the statistical analysis been performed appropriately and rigorously? 

Reviewer #1: No

Reviewer #2: Yes

Reviewer #3: N/A

3. Have the authors made all data underlying the findings in their manuscript fully available?

Reviewer #1: No

Reviewer #2: No

Reviewer #3: No

4. Is the manuscript presented in an intelligible fashion and written in standard English?

Reviewer #1: Yes

Reviewer #2: Yes

Reviewer #3: Yes

5. Review Comments to the Author

Reviewer #1: Review Comments to the Author

This is a very important topic, and it is great that the authors thought about doing this research in such a timely manner and they have made some decent justifications of why this is important in the discussion. Unfortunately, however, I believe the paper needs significant revisions before it can be considered for publication.

My main concerns are with the methodology, data collection and the results and the lack of information provided in each of these sections that make me question the validity and usefulness of the data.

Below I am outlining a detailed set of feedback for each section. Please note that his feedback is intended to support the development of the paper and preparation for future submissions.

Introduction:

Lines 49-51: The sentence starting with “Those with feeding..” seems a little out of place and context here as presented. Perhaps it can be presented as an example of a difficulty of challenging situation and include how this is more complicated by the pandemic and presence of Covid-19.

Line 54: It is unclear which type of “global Leaders” the authors are referring to. Politicians? Leaders in public health?

Lines 56-59: What do the authors hope the results to be used for? Specifying this more clearly would make it more effective. For example, in terms of an post pandemic response? Or for future pandemics? For use during this current pandemic management? All of the above?

Materials and Methods:

Participants

Line 67: references provided in two formats- needs to be fixed.

Line 70-71: Did all the families who were approached accept to participate in the study? Did any one decline participation and did the authors have to select more names?

Line 72-73: Similar to above comment, did all CCGs accept request for participation or was there any who declined?

Line 74: Who were the health professionals? What kind of clinicians/practitioners were they? And again, did all who were initially approached agree to respond?

Line 77: For additional details regarding the Kusamala+ should provide a proper title of a report or link as apposed to just a reference number.

Measures

Line 80: Trained “personal”- spelling mistake but also, trained in what? And how were they identified? How is the Catholic Medical Mission Board connected to the authors or get involved- are they were the authors are from?

Line 81: How were participants contacted and consented? How were the paper-based surveys developed and by who?

Line 83-84: Whose demographic information was asked about from the families: the respondent? the child with the disability? both?

Line 89: the CCG were asked questions in the same areas as who? The families?

Line 91: Type should be “CCGs” not “CCGS”. Also, why were the CCGs and health professionals asked for their recommendations and not the families and caregivers who actually experienced the difficulties? Or did they provide recommendations without being asked?

Line 92-93: Why were the government officials who are the furthest removed form the lived experiences of the families of children with disabilities asked to reflect on the services and the gaps and needs as opposed to the families, the CCGs or the health professionals that work directly with the families.

Analysis:

It would have been good to know which type of thematic analysis process was used to analyze the qualitative data.

Results:

I would highly recommend including the types of disabilities that were experienced by the children.

Line 107: Given the population is children, even if it seems obvious should always include the scale/units for all categories as years so not confused with months for younger kids/infants.

Line 109: When discussing family members with symptoms, it would have been useful to know it they attempted to get tested, would have liked to be tested and were not able to due to various community or family related barriers?

Line 110: Who/what are examples of trusted sources? Clarification needed.

Line 116: Figure one is meant to present the perceived negative impact of Covid-19, yet it includes improvements as well-perhaps title of table needs to be adjusted.

Line 117-118: greater discussion or examples of the results would have been beneficial as to how families were greatly impacted in the various areas identified. While examples of quotes are provided with some of the qualitative data- this is still limited and not substantial enough.

Line 122: Where referring to school closures- perhaps need to make a distinction on how this has an impact that is different from non-disabled children for the children with disabilities who actually attended school.

Line 124: Typo”3.2”. Also, it would have been good to provide a detailed break down of the numbers of the healthcare professionals and government officials and CCGs here as well for consistency with above section with families. In addition, providing demographics on this group would have been useful such as their roles/ posts, years of experience? Gender?

Line 126: Table 1- While the themes seem to be derived from all participant groups. All examples of quotes are from family members- more examples and from other participants would be of great value. Also including or acknowledging that in some cases the result may have not been what was expected.

Discussion

Line 130: While we know the pandemic will have long lasting effects- we have not yet experienced them and so the sentence may need to be slightly adjusted and the impact of course is on all, including families and children with disabilities.

Line 154: The sentence about stigma is very general and needs to be tied to the impact of the Pandemic given the focus of the paper.

Reviewer #2: This is a well written and conducted study and interesting paper dealing with a hot topic. Indeed, little is known on how COVID-19 pandemic might have affected families with children with disabilities from low to middle income countries. The present study sheds light on the perceived impacts of COVID-19 for children with disabilities and their families from the parents, community caregivers, health professionals, and government officials.

The rationale, relevance, and methodology/descriptive analyses of the study are adequately addressed. Main limitations include the small sample size and the population included in the study were only from the Kusamala+ program. I consider some adjustments to the manuscript to be required:

1. The abstract would be designed in a structured way. Some key findings are missing.

2. To better frame the context of the study and the results, it would be informative for the reader

to specify when the COVID-19 outbreak reached Zambia –especially the location of the study (Lusaka)– and

if there were any restrictive measures or localised lockdown taken by the government or the education system that affected the population of the study.

It is suggested to briefly introduce the context in a separate section. That is, first Zambia and then Lusaka should be introduced to the readers. It is also better for the authors to mention the time range of quarantine/confinements in Zambia. Also, has Lusaka province been subject to national or local restrictions?

3. Page 7, lines 135-137: The paper states that "...to our knowledge, no work has been published on the impacts of COVID-19 on children with disabilities living in low-income communities in Zambia."

The authors point out that no research has been done/published on the impacts of the COVID-19 on children with disabilities in Zambia. This sentence may not be completely correct and by searching it can be found that some studies have been done in Zambia about the impact that COVID-19 pandemic had on children with disabilities and their families. It is also better to refer to the surveys conducted in Zambia and to present its important findings and shortcomings.

DOI:

https://doi.org/10.1186/s40359-021-00583-w

https://doi.org/10.12688/hrbopenres.13077.2

4. Please report the types of disability in participants, especially given the emphasis on disabled children population.

5. Interview guide: Can you please explain the domains of questions asked, the purpose of the questions, and give some example questions?

6. About the discussion, I suggest that you could make a more in-depth discussion regarding the role of community caregivers, health professionals, and government officials.

7. It's suggested to follow a similar manner for spelling: toward/towards, among/amongst etc.

Reviewer #3: This manuscript highlights important issues families with children with disabilities in sub-Saharan Africa are struggling with due to the COVID-19 pandemic.

The data presented is very interesting, however the manuscript lacks academic rigor.

The introduction and discussion mostly consists of references to UN reports and a few academic papers. There is a need for a thorough literature review of papers already published on the impact of COVID-19 on persons with disabilities in low income countries (the Disability Inclusive Response paper by Banks et al could be a helpful starting point), and more specifically children with disabilities in African countries. Battle, Coker, Kamga, Mbazzi, Mcinney, Mupaku, Ned, Samboma to mention a few have earlier highlighted the impact COVID has had on adults and children with disabilities in sub Saharan Africa.

In the method section I would like to see more information on the survey design, e.g. who designed the survey questions, were families of children with disabilities involved in this process? What was the background of the interviewers, it is mentioned they were trained personnel, did they have any research training, were they Zambian, what language did they use to conduct the interviews. How were phone interviews recorded and who transcribed and where needed translated the recordings? There is mention of the use of Google forms, how were GDPR guidelines respected and was data kept safe? The data analysis followed some of Braun’s steps, though a full analysis plan and reference is missing. Were the findings checked with participants? What kind of referral system / care was in place for participants who shared serious health or food security concerns? It may be helpful to consult the COREQ and SRQR guidelines for qualitative study reports when revising their paper.

The text in the findings section is limited to one page. The table and figure show interesting quotes and themes. It would be interesting to describe these in more detail in the text and provide other quotes to support the themes too. The title indicates the participants are from 3 low income communities but data is not segregated by these communities. Could the authors clarify the rational of including this specifically in the title and/or explain more about the differences between the 3 communities if any in the findings section?

Similar to the introduction, the discussion section sets out important points, but is lacking references to policies and statements made around COVID-19 and disability in low income countries, and particularly sub-Saharan Africa over the past year.

I hope the authors will consider revising the manuscript, as the data give interesting insights and recommendations from a continent that is greatly affected by the pandemic, but remains underrepresented in the academic literature as well as media reports.

6. PLOS authors have the option to publish the peer review history of their article (what does this mean?). If published, this will include your full peer review and any attached files.

Reviewer #1: No

Reviewer #2: No

Reviewer #3: No

---

## [Author Response · Author response to Decision Letter 0]

5 Sep 2021

Editorial format and data+survey availability was addressed. The response to the reviewers is included in the submission as a word document.

in R2 I believe I completed administrative requests.

Sept 5: I corrected the title discrepancy.

---

## [Decision Letter · Decision Letter 1]

11 Nov 2021

Rapid health impact assessment of COVID-19 on families with children with disabilities living in low-income communities in Lusaka, Zambia

PONE-D-21-13750R1

Dear Dr. Hearst,

We’re pleased to inform you that your manuscript has been judged scientifically suitable for publication and will be formally accepted for publication once it meets all outstanding technical requirements.

Kind regards,

Maria Berghs, PhD

Academic Editor

PLOS ONE

Additional Editor Comments (optional):

A reviewer has found it odd Zambia's IRB's have exempted this study as it includes interviews / data collection with human subjects. They have requested a letter to attach to the submission records to ensure this is documented before submission.

Reviewers' comments:

Reviewer's Responses to Questions

**Comments to the Author**

1. If the authors have adequately addressed your comments raised in a previous round of review and you feel that this manuscript is now acceptable for publication, you may indicate that here to bypass the “Comments to the Author” section, enter your conflict of interest statement in the “Confidential to Editor” section, and submit your "Accept" recommendation.

Reviewer #2: All comments have been addressed

Reviewer #3: All comments have been addressed

2. Is the manuscript technically sound, and do the data support the conclusions?

Reviewer #2: Yes

Reviewer #3: (No Response)

3. Has the statistical analysis been performed appropriately and rigorously? 

Reviewer #2: N/A

Reviewer #3: (No Response)

4. Have the authors made all data underlying the findings in their manuscript fully available?

Reviewer #2: Yes

Reviewer #3: (No Response)

5. Is the manuscript presented in an intelligible fashion and written in standard English?

Reviewer #2: Yes

Reviewer #3: (No Response)

6. Review Comments to the Author

Reviewer #2: (No Response)

Reviewer #3: (No Response)

7. PLOS authors have the option to publish the peer review history of their article (what does this mean?). If published, this will include your full peer review and any attached files.

Reviewer #2: No

Reviewer #3: No

---

## [Editor Report · Acceptance letter]

6 Dec 2021

PONE-D-21-13750R1 

*Rapid health impact assessment of COVID-19 on families with children with disabilities living in low-income communities in Lusaka, Zambia*

Dear Dr. Hearst:

I'm pleased to inform you that your manuscript has been deemed suitable for publication in PLOS ONE. Congratulations! Your manuscript is now with our production department. 

Kind regards, 

on behalf of

Dr. Maria Berghs 

Academic Editor

PLOS ONE